# Evaluation of Augmented Reality Surgical Navigation in Percutaneous Endoscopic Lumbar Discectomy: Clinical Study

**DOI:** 10.3390/bioengineering10111297

**Published:** 2023-11-09

**Authors:** Xin Huang, Xiaoguang Liu, Bin Zhu, Xiangyu Hou, Bao Hai, Shuiqing Li, Dongfang Yu, Wenhao Zheng, Ranyang Li, Junjun Pan, Youjie Yao, Zailin Dai, Haijun Zeng

**Affiliations:** 1Pain Medicine Center, Peking University Third Hospital, Beijing 100191, China; hay221@163.com (X.H.);; 2Department of Orthopedics, Peking University Third Hospital, Beijing 100191, China; 3Department of Orthopedics, Beijing Friendship Hospital, Beijing 100052, China; 4State Key Laboratory of Virtual Reality Technology and Systems, Beijing Advanced Innovation Center for Biomedical Engineering, Beihang University, Beijing 100191, China; 5Smart Learning Institute, Beijing Normal University, Beijing 100875, China

**Keywords:** augmented reality, percutaneous endoscopic lumbar discectomy, real-time visualization, surgical navigation, clinical study, computer-assisted surgery

## Abstract

Background: The puncture procedure in percutaneous endoscopic lumbar discectomy (PELD) is non-visual, and the learning curve for PELD is steep. Methods: An augmented reality surgical navigation (ARSN) system was designed and utilized in PELD. The system possesses three core functionalities: augmented reality (AR) radiograph overlay, AR puncture needle real-time tracking, and AR navigation. We conducted a prospective randomized controlled trial to evaluate its feasibility and effectiveness. A total of 20 patients with lumbar disc herniation treated with PELD were analyzed. Of these, 10 patients were treated with the guidance of ARSN (ARSN group). The remaining 10 patients were treated using C-arm fluoroscopy guidance (control group). Results: The AR radiographs and AR puncture needle were successfully superimposed on the intraoperative videos. The anteroposterior and lateral AR tracking distance errors were 1.55 ± 0.17 mm and 1.78 ± 0.21 mm. The ARSN group exhibited a significant reduction in both the number of puncture attempts (2.0 ± 0.4 vs. 6.9 ± 0.5, *p* = 0.000) and the number of fluoroscopies (10.6 ± 0.9 vs. 18.5 ± 1.6, *p* = 0.000) compared with the control group. Complications were not observed in either group. Conclusions: The results indicate that the clinical application of the ARSN system in PELD is effective and feasible.

## 1. Introduction

Percutaneous endoscopic lumbar discectomy (PELD) has become a favorable treatment option for lumbar disc herniation [1]. PELD is associated with numerous advantages, including minimal injury, reliable clinical efficacy, the need for only local anesthesia, and earlier functional recovery [2]. However, the learning curve for PELD is steep [3,4], especially in terms of the aspect of puncture positioning. The approach in PELD is distinct from traditional methods: it requires a more lateral entry point [5], a longer puncture path, and a more inclined angle, making targeting more challenging. Additionally, because the skin is not cut open, surgeons cannot visualize the internal anatomical structures. At present, the puncture procedure is heavily reliant on the guidance of the C-ARM and the surgeon’s expertise. For beginners and in some difficult situations, the puncture may need to be adjusted dozens of times [6]. Excessive adjustments can lead to an increase in the number of punctures and fluoroscopic views, thereby increasing the potential risk of radiation damage [7,8]. If the puncture direction is not optimal, it might lead to obstructions by abnormal bony structures (such as high iliac crests) [9] and damage to spinal nerves [10], vessels [11], and abdominal organs [12].

To address these issues, navigation and robotics-assisted surgical techniques have been adopted in clinical practice [13,14]. However, they often necessitate an extra incision for installing fiducial markers, like spinous process bone clamps [15], which contradicts the principle of minimal invasiveness in PELD. Additionally, the necessity of obtaining intraoperative computed tomography (CT) images and performing registration procedures can prolong the surgery [16]. Furthermore, these technologies are unsuitable for primary hospitals due to the high costs and sophisticated requirements [17].

Augmented Reality (AR) is a technology that overlays virtual information in the real-world environment, offering enhanced visualization capabilities [18]. The introduction of AR technology allows surgeons to visualize the patient’s internal anatomical structures in the surgical area without moving their field of view [19]. This is particularly meaningful for PELD, which can potentially address the current challenge of non-visual puncture procedures. Moreover, the application of AR in clinical practice brings additional benefits, including flexible integration with other technologies, user-friendly features, and a low cost [20]. Presently, while there have been some studies on the application of AR in spine surgery [21], few have been applied clinically.

In this study, a novel augmented reality surgical navigation (ARSN) system has been developed. The ARSN system is equipped with three functionalities including AR radiograph overlay, real-time AR puncture needle tracking, and AR navigation. The purposes of this study were to establish the standardized ARSN navigation surgical protocols for PELD and to further evaluate the feasibility and clinical effectiveness of the ARSN system through clinical trials.

## 2. Materials and Methods

### 2.1. Patient Characteristics

The study protocol was approved by the local institutional review board (2020-378-02). All patients signed an informed consent document. Between December 2020 and March 2021, a cohort of 20 patients diagnosed with lumbar disc herniation underwent PELD treatment. The patients were randomized into two groups using a random number table. In this setup, 10 patients received treatment facilitated by the ARSN system (ARSN group), while the other 10 were treated with the conventional C-arm fluoroscopy guidance (control group). The patient characteristics are listed in Table 1.

The inclusion criteria are as follows: (1) age 18–70 years; (2) clear clinical and radiological diagnosis of lumbar disc herniation; (3) inadequate response to conservative therapy; (4) consent to PELD surgery; (5) willingness and ability to comply with the study protocol.

The exclusion criteria encompass (1) lumbar segmental instability or spondylolisthesis; (2) multi-segmental spinal canal stenosis; (3) active infection; (4) coagulation disorders; (5) pregnancy; (6) refusal to participate, as indicated by the patient.

### 2.2. System Components and Functions

The ARSN system comprises a designed hardware system and a self-developed software platform (version V1.0). The hardware system consists of an infrared positioning device (Polaris Spectra, NDI, Waterloo, Canada), two depth cameras (ZED Mini, Stereolabs, San Francisco, CA, USA), a workstation with a monitor, and a location kit (Figure 1). The location kit includes surface fiducial markers and a puncture needle locator. The system was implemented with a C-arm (Brivo OEC 715, GE, Boston, MA, USA) in the operation room.

The ARSN system possesses three functionalities: (1) AR radiograph overlay and AR visualization of spinal anatomy; (2) AR puncture needle real-time tracking; and (3) AR navigation. Prior to current clinical trials, the precision and reliability of these functionalities have been validated through animal experiments (Figure 2A). In the animal experiments, the anteroposterior (AP) and lateral errors of overlayed AR radiographs were 0.74 ± 0.21 mm and 1.13 ± 0.40 mm, respectively. The AP and lateral errors of the AR puncture needle real-time tracking were 1.26 ± 0.20 mm and 1.22 ± 0.25 mm, respectively [22]. Furthermore, to ensure that the ARSN system could be utilized effectively in clinical surgeries, phantom studies and preliminary trials for the ARSN system were conducted (Figure 2B,C). In these preliminary experiments, we successfully implemented the AR radiograph overlay and AR puncture needle real-time tracking functionalities on a phantom and a doctor. Although we did not test the accuracy in the preliminary experiment, it allowed us to gain insights into the detailed use of the ARSN system during surgeries, such as it being necessary to leave space for the C-arm between the workstation and the infrared positioning device and to ensure that the surgical area is within the center range of the workstation for tracking. In addition, through preliminary experiments, we managed to develop standardized surgical protocols for the ARSN system, thus ensuring thorough preparation for the subsequent clinical trials.

### 2.3. Standardized Surgical Protocols for the ARSN System in PELD

For the ARSN group, the puncture procedure of PELD was performed with the guidance of the ARSN system. The standardized surgical protocols for the ARSN system in PELD are shown in Figure 3.

#### 2.3.1. ARSN System Arrangement in the Operating Room

Before surgery, a detailed preoperative layout was performed to ensure the optimal positioning of all equipment. Initially, the patient’s surgical side is determined. As shown in Figure 1, the workstation of the ARSN system was placed on the opposite side of the surgical field, next to the operating table and facing the surgical field. The infrared positioning device is situated on the opposite side, adjacent to the table’s head end. The C-arm was positioned on the same side as the surgical area, located between the workstation and the infrared positioning device.

#### 2.3.2. Self-Adaptive Calibration

Considering the inconsistent positions of the infrared positioning device and the workstation, the system necessitates self-adaptive calibration before formal operation. The calibration process was conducted in the surgical area in both anteroposterior and lateral video views. First, the cameras’ video streams were activated. The fiducial markers were positioned at a designated location within the surgical area. When the recognition button was clicked, the infrared tracking device captured the three-dimensional (3D) spatial coordinates. Concurrently, the coordinates within the video scene were obtained. The 3D spatial coordinates and the coordinates within the video scene were then matched by singular value decomposition (SVD). Following the initial calibration, the fiducial markers were relocated to another spot within the surgical area, and the recognition button was clicked to initiate the second calibration. Calibration was performed at twenty distinct locations for both AP and lateral views. Ultimately, the coordinate systems of the infrared positioning device and the camera were aligned.

#### 2.3.3. AR Radiograph Capture

Following preparation, the patient entered the operating room and was positioned in a prone position on the operating table. Using a custom-made radiolucent bracket, surface fiducial markers were horizontally and perpendicularly placed on the patient’s back and flank. The surgical table, along with the patient, was then advanced into the C-arm, ensuring the patient’s surgical area was within the C-arm’s imaging range (Figure 4A). Radiographs of the lumbar spine were captured in both AP and lateral views. The positions of the C-arm and fiducial markers were adjusted until the X-ray images clearly displayed both the lumbar vertebrae and the fiducial markers (Figure 4B,C).

#### 2.3.4. AR Radiograph Overlay

The surgical table, along with the patient, was then repositioned into the ARSN system until the camera range of the ARSN system was aligned with the surgical area. At this point, the fiducial markers on the patient’s body surface can be seen in the video (Figure 5A,B). The AP and lateral AR Radiographs, in which the fiducial markers were visible, were imported into the ARSN system. The markers shown in the video and in the X-ray were identified and matched automatically, using a deep learning algorithm [22]. Consequently, AR radiographs were accurately overlaid onto the patient’s body in the video scene, achieving a precise AR radiograph overlay (Figure 5C,D). Both the intraoperative videos and the overlaid AR radiographs were shown on the monitor. The overlaid AR radiographs enabled operators to visualize the patient’s internal spinal anatomical structures and their locations.

#### 2.3.5. AR Radiograph-Guided Preoperative Localization

After the AR radiograph overlay, the surface fiducial markers were removed. The surgical target segment was identified in the AR radiographs. Utilizing the anatomical information from the AR radiographs, a Kirschner wire was positioned on the patient’s back to outline the puncture direction and identify the skin entry point. The Kirschner wire’s position was fine-tuned until its line intersected with the midpoint of the targeted intervertebral disc and the tip of the superior articular process, as seen in the AP overlaid radiograph (Figure 6A). The line along the Kirschner wire was then marked on the skin. The skin entry point was chosen and marked on this line, located 11–13 cm from the midline, varying with the patient’s body size (Figure 6B). Following preoperative localization, the surgical area underwent disinfection and draping (Figure 6C).

#### 2.3.6. AR Puncture Real-Time Tracking: External Test before Puncture

The puncture needle was affixed to a locator with markers, and its length was input into the ARSN system. This enabled the infrared tracking device to pinpoint the location of the puncture needle. Through calibration and coordinate transformation, the AR virtual puncture needle could be shown in the intraoperative video in real time. Before the puncture, an external test was conducted to validate the tracking precision. The real puncture needle was placed on the patient’s body. The location of the real puncture needle could be captured by the camera and shown in the video. Meanwhile, the AR virtual puncture needle was shown in the video. In both the AP and lateral views, the positions of the real puncture needle and the AR virtual puncture needle were recorded and compared at five different positions (Figure 7). The errors in the distance and angle were quantified utilizing the Image-Pro Plus software (version 6.0, Media Cybernetics, Rockville, MD, USA).

#### 2.3.7. AR-Navigated Puncture

Upon confirming the precision of AR tracking, the puncture procedure was performed under AR navigation. On the AR radiograph, the operator could identify the target (ventral side of the superior articular process of the surgical segment). Following local anesthesia, the puncture needle was introduced into the body through the predetermined entry point. At this time, the operator could see the position of the AR virtual puncture needle inside the body from the screen (Figure 8). Based on the position of the AR virtual puncture needle on the AR radiograph displayed by the ARSN system, the puncture needle was carefully adjusted until the AR virtual puncture needle reached the target on the AR radiograph: in the AP view, the tip of the puncture needle reached the lateral side of the superior articular process; in the lateral view, the tip of the needle reached the ventral side of the superior articular process. After the puncture, actual X-rays were taken to confirm the needle’s position, and adjustments were made, as needed.

### 2.4. Control Group

In the control group, traditional methods were employed for preoperative localization and intraoperative puncture procedures [5]. The patient was placed in a prone position. A Kirschner wire was placed on the patient’s back. Under AP fluoroscopy, the position of the Kirschner wire was adjusted until it passed through the midpoint of the targeted intervertebral disc and the tip of the superior articular process. This line of the Kirschner wire was then marked on the patient’s back. Meanwhile, a Kirschner wire was positioned obliquely from the side of the patient’s waist. Under lateral fluoroscopy, the position of the Kirschner wire was adjusted until it intersected the rear of the targeted intervertebral disc and the pars interarticularis. The resultant line from this positioning was marked on the skin. The intersection point between the two lines determined the entrance point for the needle. The puncture procedure was performed freehand under AP and lateral X-ray guidance. Multiple adjustments were made to the puncture needle until it precisely reached the desired target.

### 2.5. Outcome Measures

Following the puncture procedure, patients in both groups underwent the subsequent steps of PELD. We evaluated the number of puncture attempts and fluoroscopies and the total operation time as intraoperative outcome measures. The Visual Analogue Scale (VAS), the Oswestry Disability Index (ODI), and the complication rates were assessed both one week and one month postoperatively. At the one-month postoperative follow-up, clinical outcomes were also assessed using the modified Macnab criteria.

### 2.6. Statistical Analysis

Data were presented as the mean ± standard error (SE). The independent samples t-test and Fisher’s exact test were employed for statistical analyses. The age, BMI, number of puncture attempts, number of fluoroscopies, operation time, VAS score, and ODI score were compared using the independent samples t-test. The gender, operative level, and modified Macnab scores were compared using Fisher’s exact test. All analyses were carried out using SPSS software (version 19.0, SPSS Inc., Chicago, IL, USA). A *p*-value less than 0.05 in two-tailed tests was considered statistically significant.

## 3. Results

PELD was successfully performed on patients in both the ARSN and control groups. For the ARSN group, AR radiographs were successfully acquired and superimposed onto the body surface for all patients. The preoperative localization process was entirely accomplished using AR radiographs. The skin entrance point chosen based on the AR radiographs was accurate; no adjustments were needed during surgery. The accuracy of the AR real-time puncture needle was assessed through the external test before puncture. The AP average tracking distance error was 1.55 ± 0.17 mm, the lateral average tracking distance error was 1.78 ± 0.21 mm, the AP average tracking angle error was 1.06 ± 0.13°, and the lateral average tracking angle error was 0.88 ± 0.11°.

The comparison of intraoperative data between the two groups is listed in Table 2. The number of puncture attempts in the ARSN group was significantly lower than that in the control group (2.0 ± 0.4 vs. 6.9 ± 0.5, *p* = 0.000). The overall number of fluoroscopies in the ARSN group was significantly lower than that in the control group (10.6 ± 0.9 vs. 18.5 ± 1.6, *p* = 0.000). In the ARSN group, preoperative localization predominantly utilized the AR radiograph, eliminating the need for a C-arm. However, the process of obtaining the AR radiograph did require X-ray imaging. Thus, the number of fluoroscopies used to capture the AR radiograph was counted in the number of localization fluoroscopies. No significant differences were found in the number of localization fluoroscopies between the groups (6.6 ± 0.6 vs. 4.8 ± 0.9, *p* = 0.095). However, the ARSN group exhibited a significant reduction in the number of puncture fluoroscopies (4.0 ± 0.8 vs. 13.7 ± 1.0, *p* = 0.000) compared to the control group. There were no significant differences in the operation time between the two groups (*p* = 0.91).

The postoperative follow-up data are listed in Table 3. Patients in both groups exhibited improvements in VAS and ODI scores at 1 week and 1 month postoperatively. No statistical differences were observed in the VAS, ODI, and Modified Macnab Scores between the two groups. No complications, including nerve injury, hematoma, dural leak, or infection, were observed in either group.

## 4. Discussion

Although the PELD procedure is minimally invasive and offers various advantages, it also presents certain challenges, particularly during the puncture and localization steps. Centro Médico Teknon et al. reported that a learning curve of 72 cases was necessary to achieve the target of 90% good/excellent outcomes for PELD [6]. The challenges in the puncture and localization procedures include: (1) A more lateral approach: traditional posterior approach surgery accesses the vertebral plate directly from the back. In contrast, the skin entrance point for PELD is more lateral, approximately 9–13 cm from the midline. This special positioning increases the complexity of targeting. (2) An inability to visualize internal anatomical structures: in traditional open surgery, the skin is incised and the underlying bones are exposed, allowing surgeons to have a clear view of the internal anatomical structures. Conversely, in minimally invasive surgery, due to the absence of extensive incisions, the internal structures remain hidden. As a result, surgeons must rely on their own experience to envision the location of the internal anatomical structures. (3) Risks associated with punctures: an incorrect puncture direction may potentially injure nerve roots, blood vessels, or abdominal organs. (4) Obstruction by bony structures: an inappropriate puncture trajectory might lead to obstruction by bony structures. For instance, a high iliac crest may hinder access to the L5-S1 segment [9]. Given these complexities, the learning curve for PELD is generally steep.

Currently, the puncture procedure in PELD is conducted under the guidance of a C-arm. Surgeons can determine the position of the needle tip from both the AP and lateral radiographic views. The AP view reveals the needle tip’s distance to the midline, while the lateral view reveals the depth of needle penetration. The puncture needle is repeatedly adjusted manually based on the position in the X-ray until it reaches the target site. This process is highly dependent on the surgeon’s expertise, and for beginners, it might necessitate numerous repetitive adjustments. Frequent adjustments of the puncture may result in various issues, including increased soft tissue damage, prolonged surgical time, and potential radiation hazards from excessive fluoroscopy. Iprenburg M et al. reported that during a surgeon’s initial experience with PELD, both the patient and the surgeon are likely to be exposed to elevated levels of radiation [7].

AR technology integrates digital information into the real-world environment. Currently, it has been widely used in various fields, such as entertainment, education, and healthcare. In the realm of spinal surgery, AR is being gradually explored for its potential to assist in vertebroplasty [23], pedicle screw placement [24], and osteotomy planning [25]. One of the most significant advantages of AR in clinical practice is its ability to show the anatomical information that surgeons aim to visualize. With AR technology, surgeons can directly view the anatomical structures inside a patient’s body in real-world scenarios. This is meaningful in minimally invasive spinal surgeries, especially for PELD. During the PELD procedure, the surgeon is unable to observe the internal anatomical structures of the patient. Although the intermittent use of the C-arm can help determine the position of the puncture needle, the puncture process remains somewhat blind. Conventional navigation can reveal the anatomical structures and the positions of instruments within the patient’s body. However, surgeons usually need to divert their attention away from the surgical field to a dedicated navigation screen, which requires additional cognitive effort for thinking and analysis. The integration of AR alleviates this issue, enabling surgeons to directly see information in the surgical field without having to look away.

In this study, the ARSN system was constructed. The ARSN system possesses three functionalities: (1) AR radiograph overlay and AR visualization of spinal anatomy; (2) AR puncture needle real-time tracking; and (3) AR navigation. The overlaid AR radiograph allows the surgeon to directly observe the structure and position of the patient’s lumbar spine in the surgical area. This enables the surgeon to clearly see the target and to directly determine the puncture entry point and direction during preoperative positioning and intraoperative puncturing. We previously conducted animal experiments to verify the accuracy of the AR radiograph overlay, which can achieve a level of 1 mm [22]. In clinical trials, the reevaluation of AR radiographs’ accuracy was avoided. This was due to the anticipated increase in fluoroscopy exposure and the extended surgical durations resulting from additional experiments, which is inconsistent with the ethical standards of clinical research. In this study, the clinical effect of AR radiographs in PELD was further validated. AR radiographs were successfully acquired and superimposed onto the body surface of all patients. The preoperative positioning relied solely on these radiographs, which played a crucial guiding role during surgery.

AR real-time tracking of the puncture needle is as crucial as visualizing the patient’s AR vertebrae internally. While the needle is visible externally, its position becomes invisible to the surgeon once it penetrates the patient’s body. The AR Spinal Navigation (ARSN) system could track the needle’s position and display it on the video in real time. The tracking precision was demonstrated to be high before penetration. With the integration of AR radiograph overlay and AR needle tracking, AR navigation becomes achievable. The surgeon can adjust the AR needle in real time towards the target displayed on the AR radiograph. The ARSN system provides continuous guidance without necessitating ongoing fluoroscopy. In contrast, traditional C-Arm guidance is typically intermittent to mitigate excessive radiation exposure. Barbara Carl et al. reported the AR incorporation of 3D anatomical structures overlay in spinal microscopic surgeries but found that the superimposition of 3D structures sometimes obstructed a clear view of the surgical field [26]. Accurate depth perception stands as a challenge within the field of AR technology [27]. In our study, we employed both anteroposterior and lateral views for AR navigation. This approach effectively addressed the issue of depth perception, allowing for the accurate display of the needle’s depth. Additionally, the use of anteroposterior and lateral radiographs aligns with the conventional practices of surgeons, providing a more intuitive and familiar navigation experience.

In the study, AR guidance was used to direct the puncture needle to the area near the target, and radiographs were still required to verify and adjust the position of the puncture needle. Since PELD surgery is performed under local anesthesia, patients might shift due to pain during the operation. Additionally, the patient’s breathing can also affect the accuracy of the puncture. In this case, even if the error of AR tracking is zero, the accuracy can still be affected. Therefore, achieving a one-time successful placement is challenging. However, the initial puncture managed to reach the vicinity of the articular process, and it could be easily adjusted to the target under the guidance of the C-arm. The results indicate that the utilization of ARSN can markedly reduce the number of punctures. This not only mitigates the injuries caused by multiple punctures but also decreases the number of fluoroscopies, thereby reducing radiation exposure risks. Additionally, the application of ARSN is not complicated. The results show that the employment of ARSN does not significantly increase the duration of the surgery. To guarantee the optimal functionality of ARSN, several details should be noted: (1) the video scope of ARSN should be accurately aligned with the surgical area to ensure precision and efficiency during the operation; (2) during the AR radiograph capture, it is crucial to include both the lumbar vertebrae of the surgical segment and the markers required for registration within the X-ray image; (3) once the real space and video space are calibrated and registered, and it is essential to maintain the absolute fixation of the infrared positioning device and the workstation to ensure consistent and accurate guidance; (4) after the AR radiograph overlay, patients should be instructed to avoid any movement.

Besides reducing the number of punctures and intraoperative fluoroscopy, the results show that the application of ARSN during surgery is highly safe. Complications were not observed in either group. Moreover, patients in both groups experienced good relief one week and one month postoperatively. Ninety percent of patients in both groups achieved “excellent” or “good” recovery according to the modified Macnab criteria at the 1-month follow-up. To further enhance the safety of the ARSN system, several precautions should be taken into account: (1) during puncture, it is important to regard the AR guiding as a tool only, and the surgeon should still observe the direction of the puncture and judge the appropriateness; (2) in the course of surgery, reliance on the C-arm should not be entirely eliminated, as the C-arm can still be utilized to confirm the direction and position of the puncture needle, if needed; (3) throughout the puncture process, it is important to inquire about the patient’s lower limb sensations to monitor potential nerve damage.

Implementing the above details, the ARSN group achieved favorable results, including fewer puncture attempts and fewer fluoroscopies, while ensuring good postoperative outcomes. Nonetheless, this study has certain limitations: the modest sample size and the fact that it was conducted at a single center limit the generalizability of our findings. A multi-center study with a larger sample size is needed for further validation. The existing ARSN systems lack movement tracking and correction capabilities, which should be further developed and improved in the future.

## 5. Conclusions

In this study, we developed a novel ARSN system, equipped with functionalities including AR radiograph overlay, real-time AR puncture needle tracking, and AR navigation. The system facilitates the AR visualization of the lumbar spine structure and the puncture needle inside a patient’s body in real time, enhancing both preoperative positioning and intraoperative puncture procedures. The precision and reliability of these functionalities have been validated through animal experiments in our previous work, and the clinical effectiveness and the standardized surgical protocols for the ARSN system in PELD were further evaluated in the study. Our findings indicate that the ARSN system significantly minimizes the number of punctures and reduces the need for excessive fluoroscopy in PELD. The application of ARSN during surgery proved to be highly safe, with no observed complications and significant postoperative relief. The introduction of AR in spine navigation makes minimally invasive spinal surgeries more intuitive and visible. This technology has substantial potential. This was a preliminary study, and it should be continued. Further research such as the construction of an AR endoscope is anticipated in the future.

## Figures and Tables

**Figure 1 bioengineering-10-01297-f001:**
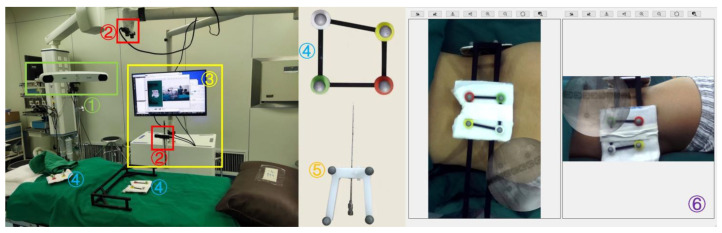
The augmented reality surgical navigation (ARSN) system is made up of (1) an infrared positioning device, (2) two depth cameras, (3) a workstation with a monitor, (4) surface fiducial markers, (5) a puncture needle with a locator, and (6) a self-developed software platform (version V1.0).

**Figure 2 bioengineering-10-01297-f002:**
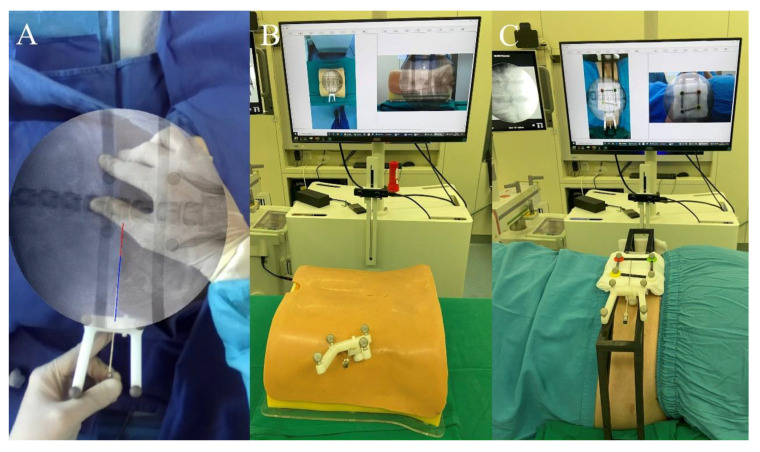
Prior to clinical trials, the precision and reliability of the ARSN system have been validated through animal experiments (**A**). Furthermore, a phantom study (**B**) and preliminary trial (**C**) were conducted to formulate standardized surgical protocols for the ARSN system.

**Figure 3 bioengineering-10-01297-f003:**
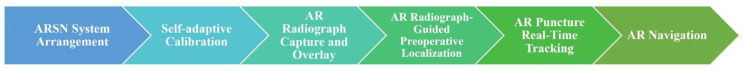
Workflow of the standardized surgical protocols for the ARSN system in percutaneous endoscopic lumbar discectomy (PELD).

**Figure 4 bioengineering-10-01297-f004:**
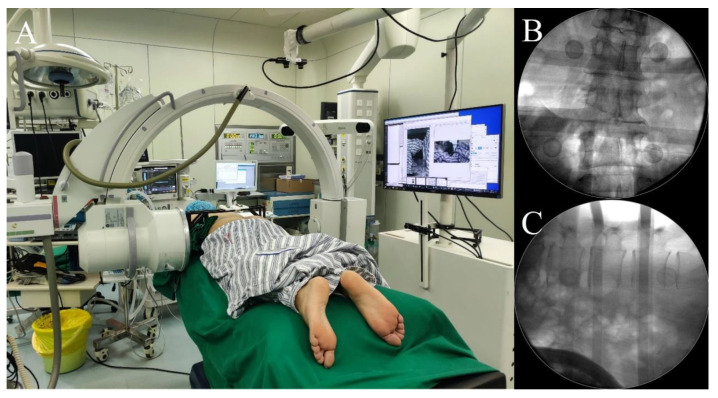
The patient’s surgical area and fiducial markers were positioned within the C-arm’s imaging range (**A**); thus, X-ray images clearly displayed both the lumbar vertebrae and the fiducial markers (**B**,**C**).

**Figure 5 bioengineering-10-01297-f005:**
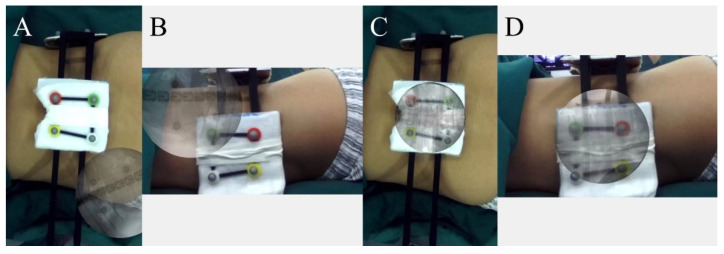
Before the overlay, the fiducial markers on the patient’s body surface could be seen in the video (**A**,**B**). After the overlay, AR radiographs were shown on the patient’s body in the video scene (**C**,**D**).

**Figure 6 bioengineering-10-01297-f006:**
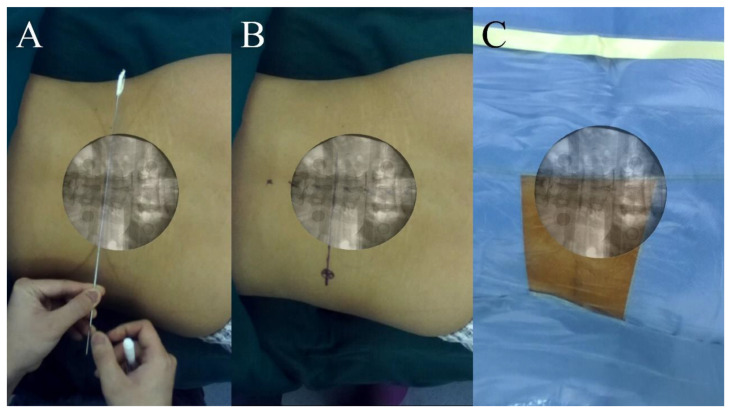
AR radiographs and a Kirschner wire were used to identify the surgical target segment and determine the puncture direction (**A**); the puncture direction and skin entry point were then identified and outlined on the patient (**B**); following preoperative localization, the surgical area was prepared through disinfection and draping (**C**).

**Figure 7 bioengineering-10-01297-f007:**
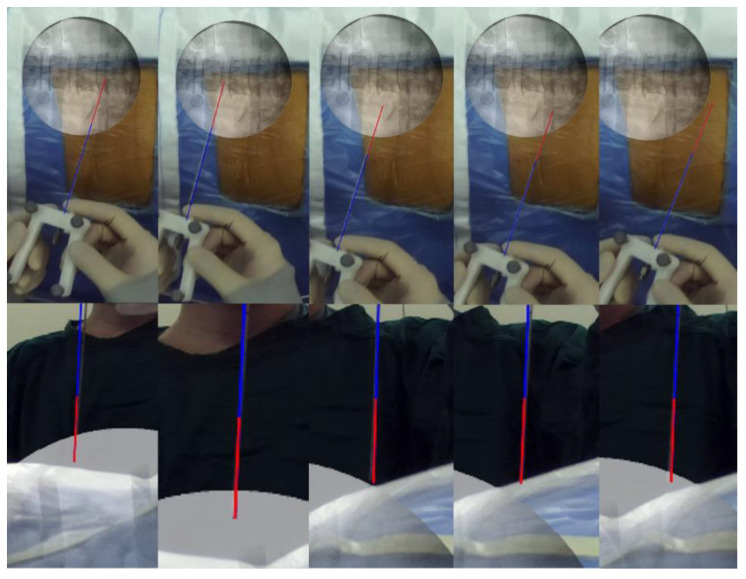
Before the puncture, the positions of the real puncture needle and the AR virtual puncture needle were recorded and compared at five different positions to validate the tracking precision.

**Figure 8 bioengineering-10-01297-f008:**
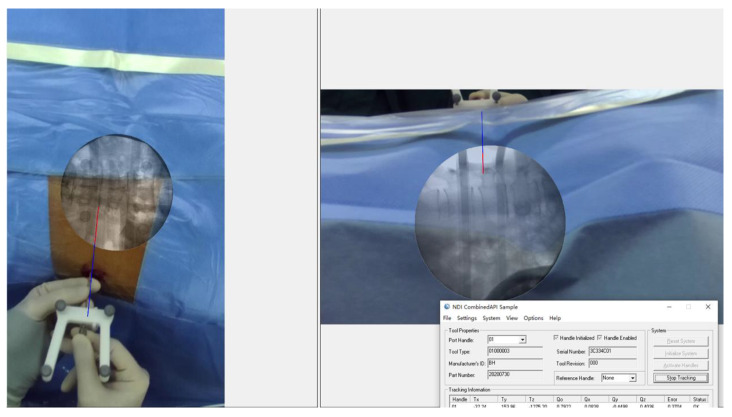
The puncture procedure was executed under AR navigation, allowing the operator to visualize both the AR virtual puncture needle within the body and the target on the AR radiograph.

**Table 1 bioengineering-10-01297-t001:** Patient characteristics.

Characteristics	ARSN Group	Control Group	*p*-Values
Age (years)	46.9 ± 3.7	43.5 ± 4.7	0.57
Gender			0.5
Male	5	6	1
Female	5	4	
BMI (kg/m^2^)	24.3 ± 0.5	25.6 ± 1.0	0.26
Operative level			1
L2/3	0	1	
L3/4	1	1	
L4/5	4	3	
L5/S1	5	5	

ARSN augmented reality surgical navigation.

**Table 2 bioengineering-10-01297-t002:** Comparison of intraoperative data between the two groups.

Characteristics	ARSN Group	Control Group	*p*-Values
Number of Puncture Attempts	2.0 ± 0.4	6.9 ± 0.5	0.000
Overall Number of Fluoroscopies	10.6 ± 0.9	18.5 ± 1.6	0.000
- AP Fluoroscopies	5.2 ± 0.6	9.1 ± 0.8	0.001
- Lateral Fluoroscopies	5.4 ± 0.6	9.4 ± 0.8	0.001
Number of Localization Fluoroscopies	6.6 ± 0.6	4.8 ± 0.9	0.095
- AP Fluoroscopies	3.2 ± 0.5	2.2 ± 0.4	0.117
- Lateral Fluoroscopies	3.4 ± 0.3	2.6 ± 0.5	0.202
Number of Puncture Fluoroscopies	4.0 ± 0.8	13.7 ± 1.0	0.000
- AP Fluoroscopies	2.0 ± 0.4	6.9 ± 0.5	0.000
- Lateral Fluoroscopies	2.0 ± 0.4	6.8 ± 0.5	0.000
Operation Time (min)	75.8 ± 11.2	74.3 ± 7.2	0.911

**Table 3 bioengineering-10-01297-t003:** Comparison of Clinical Outcomes between the ARSN Group and Control Group.

Characteristics/Metrics	ARSN Group	Control Group	*p*-Values
Preoperative Metrics:			
VAS	7.3 ± 0.3	7.8 ± 0.2	0.247
ODI	31.0 ± 1.8	29.5 ± 2.8	0.661
Post-Surgery Metrics (1 Week):			
VAS	1.3 ± 0.4	1.4 ± 0.5	0.866
ODI	8.2 ± 1.9	9.3 ± 2.4	0.724
Post-Surgery Metrics (1 Month):			
VAS	0.8 ± 0.3	1.1 ± 0.3	0.517
ODI	6.6 ± 1.4	7.9 ± 1.5	0.532
Complication Rate	0	0	1
Modified Macnab Scores			1
Excellent	5	4	
Good	4	5	
Fair	1	1	
Poor	0	0	

## Data Availability

Not applicable.

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
