# Peer review of "Evaluation of Augmented Reality Surgical Navigation in Percutaneous Endoscopic Lumbar Discectomy: Clinical Study"

_bioengineering, 2023, doi:10.3390/bioengineering10111297_

Round 1
Reviewer 1 Report
Comments and Suggestions for Authors
This work designs an augmented reality surgical navigation system for puncture procedures during percutaneous endoscopic lumbar discectomy. This work seems to have been carefully completed and gave some detailed results. However, the manuscript still contains many minor points and needs to be revised before it is finally published.
1. Abbreviation is needed before the Introduction section.
2. All abbreviations should be defined in full the first time they appear in the title, abstract, main text, and figure or table captions, even if they are well known in the field.
3. The author's introduction needs to be optimized, and we suggest that the author evaluate what needs to be improved in the introduction according to the following criteria: What is the problem to be solved? Are there any existing solutions? Which is the best? What is the main limitation of the best and existing approaches? What do you hope to change or propose to make it better? How is the paper structured?
4. It is recommended that authors provide the interface of the designed software.
5. The conclusion of this paper needs to be optimized. The authors may give the details of their paper's novelty with short descriptions. It is suggested that the author add some comparisons with previous work, advantages and disadvantages of the author's method, and prospects for future research directions.
Reviewer 2 Report
Comments and Suggestions for Authors
I would like to extend my heartfelt congratulations to the authors for their groundbreaking work in this area. Their innovative approach to incorporating augmented reality into spinal surgery is commendable and sets a new benchmark for others in the field.
Looking forward, while the current work has laid an impressive foundation, it would be intriguing to see further research perhaps directed towards the construction of a virtual endoscope. Such advancements could potentially revolutionize the way surgeries are performed, offering even greater precision and visualization capabilities. Keep up the excellent work, and I eagerly anticipate future contributions from this talented team.
Author Response
Thank you very much for taking the time to review this manuscript. We appreciate your recognition of our work and manuscript. Your encouragement has given us the strength and courage to keep moving forward. Although the research process had its challenges, we will continue to work hard, further strengthening the research on AR in the medical field, especially the construction of a virtual endoscope. Thank you for your suggestions! We hope to have more opportunities in the future to engage in deeper discussions with you.
Reviewer 3 Report
Comments and Suggestions for Authors
This is a manuscript about a clinical study regarding a navigation system used in PELD. It is an important and interesting study that has future probable application in a clinical setting. It is a publishable work that is very well written and structured, only needing some minor revisions.
- Line 56: ARSN is an abbreviation that has to be written the first time mentioned before it is cited in the manuscript. It is already in the abstract but it should be in the introduction.
- Line 57: At the end of the introduction, the aim has to be here and clear.
- Line 74: p-values regarding comparison between groups should be in the results section and not in material and methods.
- Line 230: Regarding the outcome measures, it would have been interesting to have different observers evaluating these scales, to have an interobserver disagreement score.
- Line 253: What was the statistical test used for this comparison, it should be clear when presenting all the p-values of the results section.
- Conclusion: The authors should state that this was a preliminary study and that it should be continued.
Reviewer 4 Report
Comments and Suggestions for Authors
I greatly congratulate the authors for their prospective randomized controlled trial, collecting 20 patients with lumbar disc herniation treated with percutaneous endoscopic lumbar discectomy (PELD), to investigate the feasibility and effectiveness of the augmented reality surgical navigation (ARSN) system.
I think the technical contents (Background, Methods, Results, Discussion, and Conclusions) of the submitted manuscript are well written and summarized.
The authors reveal that significantly minimizing the number of punctures and reducing the need for excessive fluoroscopy in PELD, ARSN during minimally invasive spinal surgeries has been proven to be highly safe.
Based on the authors’ study, I strongly agree with the idea that ARSN system has substantial potential and is worth further application in the future.
Therefore, the importance of the study presented is quite high.
I do not have any further substantial amendments to suggest.
Thank you.
Author Response
Thank you very much for taking the time to review this manuscript. We appreciate your recognition of our work and manuscript. Your encouragement has given us the strength and courage to keep moving forward. Although the research process had its challenges, we will continue to work hard, further strengthening the research on augmented reality in the medical field.